# Multidisciplinary Intensive Rehabilitation Program for People with Parkinson’s Disease: Gaps between the Clinic and Real-World Mobility

**DOI:** 10.3390/ijerph20053806

**Published:** 2023-02-21

**Authors:** Moriya Cohen, Talia Herman, Natalie Ganz, Inbal Badichi, Tanya Gurevich, Jeffrey M. Hausdorff

**Affiliations:** 1Center for the Study of Movement, Cognition and Mobility, Neurological Institute, Tel Aviv Sourasky Medical Center, Tel Aviv 6492416, Israel; 2Ezra Lemarpeh Center, Bnei Brak 5111501, Israel; 3Movement Disorders Unit, Neurological Institute, Tel Aviv Sourasky Medical Center, Tel Aviv 6492416, Israel; 4Sagol School of Neuroscience, Sackler Faculty of Medicine, Tel Aviv University, Tel Aviv 6997801, Israel; 5Department of Neurology, Sackler Faculty of Medicine, Tel Aviv University, Tel Aviv 6997801, Israel; 6Department of Physical Therapy, Sackler Faculty of Medicine, Tel Aviv University, Tel Aviv 6997801, Israel; 7Rush Alzheimer’s Disease Center and Department of Orthopedic Surgery, Rush University Medical Center, Chicago, IL 60612, USA

**Keywords:** Parkinson’s disease, gait, balance, self-management, wearables, multidisciplinary rehabilitation

## Abstract

Intensive rehabilitation programs improve motor and non-motor symptoms in people with Parkinson’s disease (PD), however, it is not known whether transfer to daily-living walking occurs. The effects of multidisciplinary-intensive-outpatient rehabilitation (MIOR) on gait and balance in the clinic and on everyday walking were examined. Forty-six (46) people with PD were evaluated before and after the intensive program. A 3D accelerometer placed on the lower back measured daily-living walking during the week before and after the intervention. Participants were also stratified into “responders” and “non-responders” based on daily-living-step-counts. After the intervention, gait and balance significantly improved, e.g., MiniBest scores (*p* < 0.001), dual-task gait speed increased (*p* = 0.016) and 6-minute walk distance increased (*p* < 0.001). Many improvements persisted after 3 months. In contrast, daily-living number of steps and gait quality features did not change in response to the intervention (*p* > 0.1). Only among the “responders”, a significant increase in daily-living number of steps was found (*p* < 0.001). These findings demonstrate that in people with PD improvements in the clinic do not necessarily carry over to daily-living walking. In a select group of people with PD, it is possible to ameliorate daily-living walking quality, potentially also reducing fall risk. Nevertheless, we speculate that self-management in people with PD is relatively poor; therefore, to maintain health and everyday walking abilities, actions such as long-term engaging in physical activity and preserving mobility may be needed.

## 1. Introduction

Exercise and physical training can help to reduce the impact of many of the motor and non-motor symptoms that negatively impact older adults with Parkinson’s disease (PD). Multidisciplinary intensive rehabilitation treatments (MIRT) have become increasingly popular, leveraging therapies from different disciplines to alleviate multiple symptoms. For example, Frazzitta and colleagues investigated the effects of MIRT, compared to medication treatment only, in individuals with early PD [1]. Compared to controls, motor symptoms and mobility improved in the MIRT group, immediately post-intervention and 3 months after completion of the intervention. Interestingly, in the intervention group, there was no change in Levodopa Equivalent Daily Dose (LEDD) for about two years later, in contrast to an increase in the control group. This contrast suggests that MIRTs may have profound, positive effects on PD symptoms and disease progression [2,3]. Similarly, Cohen et al. [4] demonstrated the feasibility of a multidisciplinary, intensive, *outpatient* rehabilitation (MIOR) program for moderate to advanced patients (Hoehn and Yahr stage ≥ 2). Significant improvements in the number of falls, functional independence, quality of life, balance, upper limb function, and reading vocal intensity were observed. Recently, 143 advanced people with PD completed a 3-week day-clinic multidisciplinary rehabilitation, and the effects on motor and non-motor symptoms were studied [5]. The results suggest that even a short-term outpatient daycare approach can positively influence motor and non-motor symptoms as well as health-related quality of life in people with PD. Aerobic and resistance exercise in general [6,7,8,9,10], and MIRTs, clearly have a positive influence on multiple PD aspects and symptoms. Nonetheless, it is still not known if the effects of a multidisciplinary intensive outpatient rehabilitation (MIOR) program carry over to impact the quantity and quality of real-world, daily-living walking.

Mobility refers to the physical movement of individuals as they experience the world. Daily-living mobility has been described as the everyday spatiotemporal patterns of an individual’s movement in their environment [11,12,13]. While in the past, self-report has often been used to evaluate daily-living walking, as part of mobility, more recent work uses wearable sensors for this purpose in healthy older adults and in people with PD [14,15,16,17,18,19].

Several papers have described interventions that were designed to improve daily-living step counts [20,21]. However, less is known about how the spatial-temporal parameters of gait measured during daily-living change in response to an intervention. Accumulating work suggests that gait and balance, as measured using conventional testing in clinical and laboratory settings, do not necessarily reflect the parallel measures during daily-living [22,23,24,25,26,27]. Giannouli et al. [22] demonstrated only mild to moderate correlations between lab-based measures of spatial-temporal measures of gait and daily-living measures. Multiple regression analyses indicated that the laboratory measures accounted for a significant but very low proportion of the variance (between 5% and 21%) of the corresponding daily-living measures. Similarly, Hillel et al. [23] showed that in older adults, the average daily-living gait speed is similar to that of dual-task gait speed in the lab. In addition, Shema-Shiratzky et al. [24] reported that the typical values of daily-living gait speed and cadence were significantly lower than usual-walking values in the lab in both patients with multiple sclerosis and age-matched healthy controls. Finally, Warmerdam et al. summarized studies that directly compared assessments made in unsupervised and supervised (e.g., in the laboratory or hospital) settings, pointing to large disparities, even in the same parameters of gait and mobility. These differences appear to be affected by psychological, physiological, pathological, cognitive, environmental, and technical factors. To facilitate the successful adaptation of the unsupervised assessment of daily-living gait and mobility into clinical practice and clinical trials, clinicians and researchers should consider these disparities and the multiple factors that contribute to them.

With that in mind, the aims of the present study were: (1) to confirm that a multidisciplinary intensive outpatient rehabilitation improves measures of gait, balance, and other PD symptoms when people with PD are tested in a clinical setting; and (2) to evaluate, for the first time, the influence of MIOR on objective measures of the quantity (e.g., step counts) and quality (e.g., spatial-temporal measures) of daily-living walking. 

Based on the previous work using MIRTs in PD, we hypothesized that gait, balance, and related symptoms would improve during conventional testing as well as during daily living. As a secondary aim, we examined retention effects in a subset of subjects, re-evaluating those subjects three months after the completion of the program. In addition, in exploratory analyses, we examined the daily-living changes that occurred in subjects whose daily-living step counts increased in response to the intervention.

## 2. Materials and Methods

This was an open-label study, pre-post intervention with a 3-month follow-up. Fifty-one older adults with PD who ranged in age from 51 to 85 years old were recruited from a multidisciplinary intensive outpatient rehabilitation (MIOR) program at the “Ezra LeMarpe” rehabilitation center in Bnei Brak, Israel. The study was conducted in collaboration with the Parkinson’s and Movement Disorders Unit at the Tel Aviv Sourasky Medical Center and was approved by the local Helsinki Committee.

**Inclusion criteria included**: idiopathic PD diagnosis as defined by the Movement Disorders Society, Hoehn and Yahr stage 2 to 4, stable anti-parkinsonian medication regimen for at least one month prior to the study, expected to remain on a stable medication regimen until the 3-month follow-up, Montreal Cognitive Assessment (MoCA) ≥ 14 and the ability to walk independently for at least 5 min with or without an assistive device. 

**Exclusion criteria included**: people with any neurological condition other than PD (e.g., stroke, MSA, Parkinsonism, PSP) or orthopedic disease which may impair normal gait and balance were excluded. In addition, we excluded people with any medical, surgical, or psychiatric conditions that might prevent subjects from complying with all aspects of the intervention.

The study subjects participated in 8 weeks of the MIOR, 3 times a week, 5 h per day. The program was comprised of physical therapy, occupational therapy, speech therapy, hydrotherapy, dance, and boxing (see details below). The broad clinical evaluation was performed three times: before, immediately after the intervention (within 1 week), and 3 months later. 

### 2.1. The Multidisciplinary Intensive Outpatient Rehabilitation (MIOR) Program

**Physical therapy:** three times a week subjects practiced different functional tasks, e.g., transitions such as sit-to-stand and getting up and down from the floor, getting in and out of bed, and transferring into and out of a car. Subjects also practiced static and dynamic balance training in several levels of difficulty and different gait training: Nordic sticks walking, indoor and outdoor walking, walking on different surfaces, and walking between obstacles. Once a week, subjects attended drumming sessions with physio balls while sitting and standing to improve their range of motion, rhythmicity, strength, and aerobic capacity. Additionally, subjects were asked to create a weekly schedule that includes performing daily exercise even after the end of the program. 

**Occupational therapy:** at each session, subjects practiced a variety of daily functions, e.g., dressing, transfers, cooking, and showering. First, motor and cognitive function were trained, then motor sequences required to perform the task were trained and finally, the task itself was practiced. Subjects practiced strategies to cope with the disease difficulties and to become familiarized with appropriate aids.

**Speech therapy:** the sessions aimed to improve speech comprehension. The training included exercises based on the Lee Silverman Voice Treatment (LSVT) method [28], multiple repetitions of exercises performed aloud, and a gradual increase in the difficulty of the practice-first creating single words, later sentences, reading passages and paragraphs, and eventually spontaneous speech. Additional individualized treatment was given if necessary to patients with low speech intelligibility, difficulty in speech fluency, and significantly low volume. Participants with swallowing difficulties also attended a group therapy session that was conducted once a week and involved strengthening the muscles around the mouth and throat, identifying an appropriate swallowing position, learning suitable techniques for safe swallowing, and instruction on proper food and drink consumption for each patient. 

**Hydrotherapy:** once a week subjects had individual aquatic treatment for about 30 min, matching the personal and specific goals set at the beginning of the treatment. It started with exposure to the water as a safe environment for stretching, muscle strengthening, performing movements at high amplitude, and improving balance. Swimming was encouraged as an aerobic exercise. 

**Dance:** once a week participants had a group practice under the guidance of a movement therapist or a certified dance teacher. They started in sitting accompanied by quiet music and later in standing accompanied by rhythmic music. Emphasis was placed on performing movements with a large amplitude, practicing balance while standing, e.g., standing on one leg, training memory of movement sequences, creating communication between the group participants by practicing in pairs, and imitating dance movements of the other participants. 

**Boxing:** subjects practice once a week under the guidance of a boxing coach. First, they learned the types of punches in standing when used with pillows, later on a combination of several punches, and finally, practicing defense movements. Emphasis was placed on the importance of hand-eye contact, movement coordination, and improving balance while standing and walking.

The participants were divided into small groups (each of 4–6 subjects) according to their cognitive state (via the MoCA score), level of independence and level of speech comprehension. During every day of the program, they participated in physical therapy, occupational therapy and speech therapy that lasted about 50 min each. Additionally, as mentioned above, the subjects also participated in boxing, dance and hydrotherapy as well as once a week an emotional-affective treatment that was run by a social worker. All participants received similar instructions regarding exercise and were referred to centers in the community. The trainers, both the PT and the OT, encourage the participants to maintain physical activity, but no structured protocol for self-exercises was delivered.

### 2.2. Clinical Evaluation in the Lab

Motor function was assessed via several widely used tests. The Timed up and Go (TUG) test quantified gait and functional mobility. The Mini Balance Test (MiniBeST) evaluated four different aspects of the balance system, and the Movement Disorders Society Unified Parkinson’s Disease Rating Scale, motor part (MDS-UPDRS part III) performed by a trained rater, was used to measure the severity of parkinsonian motor symptoms. Walking abilities and endurance were tested using the six-minute walking test (6MWT) and the ten-meter walk test (10MWT). Freezing of gait was assessed using the New Freezing of Gait Questionnaire (NFOG-Q) and for functional mobility and strength, we used the 30 s sit-to-stand test. 

Cognitive function was assessed by a trained rater using the Montreal Cognitive Assessment (MoCA). The Falls Efficacy Scale International (FES-I), the Geriatric Depression Scale (GDS), and the SF-12 health survey were administered to evaluate non-motor symptoms. The Levodopa Equivalent Daily Dose (LEDD) was calculated for each participant according to established methods [29]. 

### 2.3. Assessment of Daily-Living Walking 

A tri-axial accelerometer (Axivity AX3, York, UK, 23.0 × 32.5 × 7.6 mm^3^, 11 g, water-resistant, sampling rate: 100 Hz) was adhered to the participant’s lower back using a piece of Opsite Flexifix transparent adhesive skin tape for 7 days. The data were processed using Matlab software (MATLAB Release 2018b, The MathWorks, Inc., Natick, MA, USA). A number of studies have examined the properties and the validity of the measures derived from a lower back 3-D accelerometer [14,15,16,18,24]. In addition, Del Din et al. showed that measures based on a lower-back accelerometer were similar when they were examined at multiple time points in older adults and patients with Parkinson’s disease [20].

Quantitative (the amount of walking, for example, step counts) and qualitative (e.g., spatial-temporal measures of gait) information on the subject’s daily-living walking were derived from the recorded signals using previously described methods [17,18]. See also in the supplementary material. Only recordings with at least 3 full days of monitoring were included in the analyses. Daily-living quantity measures were comprised of daily-living step count, the percentage of the day the subject is active, and mean signal vector magnitude (SVM), a widely used general measure of overall physical activity [17,30,31,32].
$$\mathrm{SVM}=\sqrt{V^{2}+{\mathrm{ML}}^{2}+{\left(\mathrm{AP}\right)}^{2}}$$
where V refers to vertical, ML medio/lateral, and AP antero/posterior directions.

### 2.4. Qualitative Measurements of Daily-Living Gait

Cadence—the number of steps per minute, has been suggested as an approach that may be useful in assessing compliance with current physical activity guidelines. Although it still relies on a step count, it emphasizes the step rate, thereby acting as a method to estimate the intensity [33]. The normal range for usual walking is 100–115 steps per minute.

Step length [cm]—estimated step length, for example, from the heel strike of the right foot to the heel strike of the left foot. The average step length in a healthy population is 50–100 cm. Many previous studies have shown that step length is reduced among people with PD, compared to healthy controls, when subjects are evaluated in a controlled, clinical setting. The daily median value of the index was calculated and averaged over all days.

Gait speed [m/s]—step length/step time. The daily median value was calculated and averaged over all days. The average walking speed in a healthy population ranges from 0.80 to 1.60 m per second. A reduced walking speed, when measured in the lab or clinic, has been well-documented. It may explain many of the gait alterations in PD. 

Stride and step regularity (nu) in the vertical plane, tested in walking bouts of 30 s or longer. Lower values of regularity represent greater gait asymmetry; higher values of stride regularity reflect greater stride-to-stride consistency. Further details are in the supplementary material.

### 2.5. Statistical Analyses

Descriptive statistics are reported as number (percent) for nominal variables, median (quartile range) for ordinal variables, and as mean ± standard deviation (SD) for continuous variables. Median and interquartile range are presented for continuous variables if the distribution was not normal, as assessed by the Kolmagorov-Smirnof statistic. Pre-post comparisons and *p*-value are based on paired *t*-tests (parametric or non-parametric, as appropriate). Statistical analyses were performed using SPSS version 26.0 (IBM, Chicago, IL, USA). A *p*-value less than or equal to 0.05 was considered statistically significant.

## 3. Results

Fifty-one individuals with PD agreed to participate in the program, while data analysis was conducted on 46 subjects. Five subjects were excluded from the statistical analysis because of various reasons, see Figure 1. 

Participant characteristics and demographics are depicted in Table 1. The mean age and disease duration were 70.1 and 9.2 years, respectively. About two thirds of the participants were in stage 2 and 2.5 according to the “ON” Hoehn & Yahr scale and 32.6% were women. 

The follow-up testing 3 months after the completion of the intervention was accomplished by 32 participants; altogether, 14 people with PD did not complete the follow-up due to several reasons, please see Figure 1. Hoehn and Yahr (H & Y) staging at baseline was similar (*p* = 0.44) in subjects who completed the 3-month follow-up (H & Y2: n = 16; 66.7%; H & Y: n = 3; 2.5: 12.5%; H & Y3: n = 4, 16.7%, H & Y4: n = 1, 4.2%) and those who did not.

The effects of the Intervention on conventional in-lab clinical tests are summarized in Table 2. Many measures improved after the intervention. These included the MiniBEST, 6MWT, and both the TUG and 10MWT under single and dual tasking. In contrast, the MDS-UPDRS part III, LEDD, and the number of falls per month did not change. Interestingly, in terms of motor function, many improvements persisted at follow-up when comparing the values 3 months after completion of the rehab program to baseline values (n = 32), see Table 3. Significantly better scores were found in the NFOG-Q, the MiniBEST, 6MWT, and both the TUG and 10MWT under single and dual tasking at 3 months. Conversely, the improvements observed immediately after the program in sit-to-stand, MoCA, and GDS were not maintained and returned to baseline levels. To note, there were no differences in key characteristics at baseline (e.g., age, sex, disease duration, UPDRS part 3, Hoehn and Yahr stage, MOCA, LEDD, gait speed, and Timed Up and Go) in those who completed the 3-months follow-up and those who did not (*p* > 0.15). 

Since our cohort was comprised of relatively older and advanced people with PD, we performed a Spearman’s correlation analysis of disease duration and the pre-post change in gait speed, step count, timed up and go test, and the MDS-UPDRS part 3. Only the pre/post difference in the MiniBest was mildly correlated with age (Rs = 0.292; *p* = 0.049). No other significant correlations were found (*p* > 0.127). 

All of the measures of daily-living physical activity, daily-living gait quantity, and daily-living gait quality did not improve in response to the intervention (*p* > 0.14) (see Appendix A). None of these measures were significantly different from baseline levels at follow-up testing 3 months after completion of the intervention, suggesting no improvement and no deterioration. In secondary analyses, we aimed to see if the adjustment for baseline characteristics might impact daily-living measures. After adjusting for age, sex, disease characteristics, and baseline measures, all of the effects of the intervention on daily-living measures remained non-significant (*p* > 0.05). 

In an exploratory analysis, we divided the subjects based on those in whom daily-living step counts increased after the intervention (responders) compared to those who did not (non-responders). The average number of daily-living steps post-intervention was compared to pre-intervention steps in the responders and non-responders (Figure 2). While the group of responders (n = 15) increased the daily number of steps by an average of 1602 steps per day after the intervention, the non-responders (n = 23) reduced the average daily number of steps (*p* < 0.001). In the responders, step regularity improved (from: 0.48 ± 0.20 to: 0.46 ± 0.21, *p* = 0.038) as did stride regularity (from: 0.43 ± 0.19 to: 0.40 ± 0.21, *p* = 0.022); these measures did not change in the non-responders (*p* > 0.720).

## 4. Discussion

Immediately after the MIOR intervention, multiple aspects of gait and balance that were measured in the clinical setting improved. Several non-motor symptoms also responded to the intervention. Moreover, many of these improvements remained better than the baseline, pre-intervention values even 3 months later, suggesting some long-term retention. These positive benefits in the clinic are in line with previously reported studies [1,3,5]. In contrast, changes (or even positive trends) were not seen in any of the daily-living measures. Daily-living walking, i.e., gait quantity and daily-living gait quality, and the number of falls did not improve after the MIOR. In regards to the responders, who increased their daily number of steps after the intervention, perhaps they followed the instructions for daily exercise more carefully than the non-responders. 

Additionally, the participant’s older age and relatively long disease duration may have influenced our findings. The mean age of the participants was 71 years old and they had relatively advanced PD. Nevertheless, as noted above, only the change in the MiniBest after the intervention was mildly correlated with age. Since the change in the MiniBest score was mildly related to age and the present cohort had relatively advanced disease, in the future, it would be interesting to investigate the effects of the MIOR on subjects with shorter disease duration and more mild disease severity.

The disparity between the effects of the MIOR on clinical outcomes compared to daily-living measures is consistent with previous observational studies that indicated that in-clinic gait and balance do not necessarily reflect actual “functional” mobility during daily-living [22,25,34,35,36]. This could be explained by several factors. In the clinic, subjects may exert more effort or pay more attention to their gait, thus displaying better performance in the presence of a clinician (reverse white coat syndrome) compared to routine function at home. In addition, daily-living function may be affected by ingrained behaviors (e.g., sleep-wake cycles) that were not targeted by the MIOR in the present study. It also may help to keep in mind that while many measures of gait and balance did improve in the clinic and the dual-task gait speed improved by more than 0.05 m/s, i.e., more than the minimal clinically significant difference, general disease severity as measured by LEDD and MDS-UPDRS part III did not improve. Perhaps these aspects of disease severity have more of an impact on daily-living physical activity and gait than clinic-based performance measures. In the future, it would be interesting to see if an intervention that did improve these key signs of disease severity can also improve daily-living measures of gait quality and gait quantity. Regardless of the exact mechanisms, these findings underscore the importance of distinguishing between motor performance in the clinical setting, i.e., a relatively sterile environment, and actual performance during daily-living [37].

The present results suggest that to improve daily-living walking in older adults with PD, an MIOR program that improves in-clinic measures of gait and balance is not sufficient. Perhaps, a more intensive intervention possibly, with larger effects or longer duration, might be needed. An additional explanation may be that improvement in the functional ability seen during the performance of specific tasks in the clinic does not translate into daily living activities automatically. Special attention may be needed by health professionals to facilitate the enhancement of self-management; empowerment of patients through holistic care and being person-centered, maximizing motivation and capability for the patients and the importance of empowerment of caregivers to support self-management [38]. It is possible that educational sessions to the patient and family/caregiver that emphasize the need to change a sedentary lifestyle, combined with biofeedback, may lead to better daily-living mobility and a form of targeted ‘transitional” occupation therapy need to be delivered during the last period of MIOR to encourage and bring about effective daily-living changes.

Although previous studies have shown that engagement in self-management behaviors improves health and well-being in PD, the literature on intervention protocols, designated assessment tools, and the benefits of participating in self-management in patients with neurological conditions is limited. In a systemic review from 2020 [39], seven main themes were derived concerning the self-management of people with PD, including physical exercise, self-monitoring techniques, and maintaining independence. More recently, the same group [40] has focused on three main themes including the management of physical symptoms, i.e., engaging in physical activities, adapting lifestyles, managing medication, and using e-health resources.

## 5. Conclusions

The MIOR program delivered to people with PD clearly improved motor and non-motor symptoms in-lab testing, however, it is not yet clear if improvements in self-management can influence daily-living physical activity, more specifically, gait quantity or quality. The current findings suggest that perhaps using new technologies to monitor the type, quantity, and quality of daily activities may assist in the planning and development of meaningful and more effective interventions for people with PD [22,34,35,37,41]. Approaches for translating the clinical improvements that the MIOR produced into daily-living activities, possibly by leveraging tailored self-management tools, need to be further investigated to move beyond the clinic and influence daily-living activity. Research should build on these findings to develop acceptable and effective self-management tools for use in practice with people affected by PD.

## Figures and Tables

**Figure 1 ijerph-20-03806-f001:**
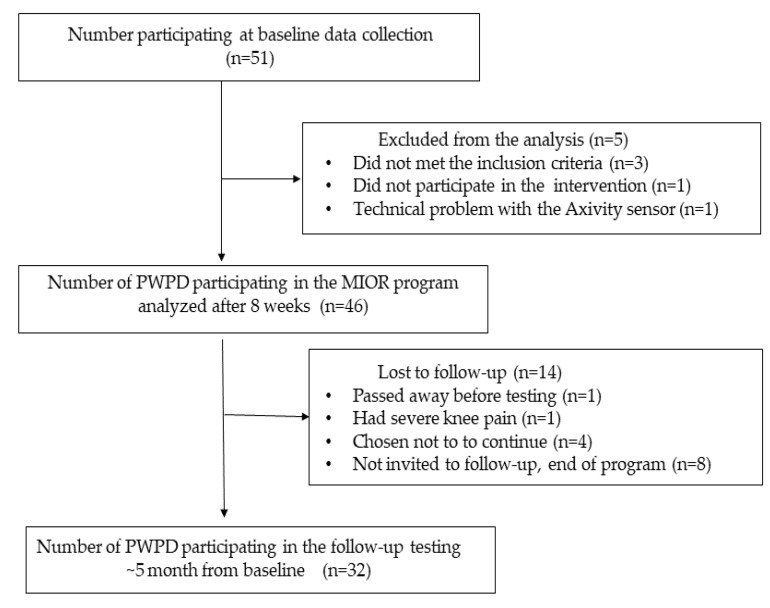
Study flowchart of participants at baseline, immediate, and 3 months after completion of the MIOR program.

**Figure 2 ijerph-20-03806-f002:**
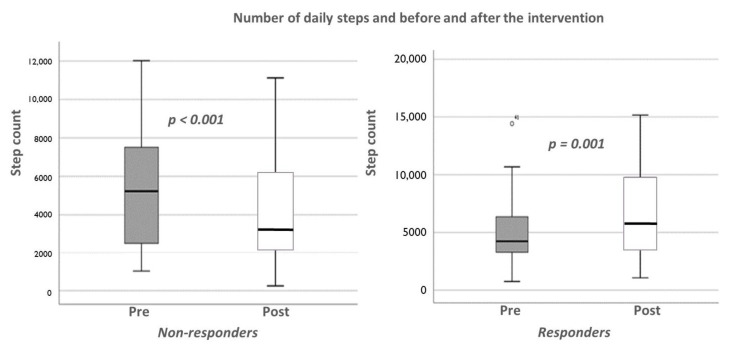
The average number of daily-living steps post-intervention as compared to pre-intervention in the responders and non-responders. The group of responders (n = 15) increased the daily number of steps by an average of 1602 steps per day after the intervention (in grey); the non-responders (n = 23), in white, reduced the average daily number of steps (*p* < 0.001). In the responders, one subject was an outlier in the number of daily steps but other basic characteristics were similar to the responder’s group pre intervention. Note that the range for the *y*-axis differs in the graph for the responders and non-responders.

**Table 1 ijerph-20-03806-t001:** Participant characteristics.

Variable	n = 46
Age (years)	70.1 ± 7.8
Gender (n, % male)	31 (67.4)
Height (cm)	168.8 ± 8.1
Weight (kg)	73.83 ± 13.99
BMI (kg/m^2^)	25.8 ± 3.9
Disease duration (years)	9.2 ± 6.2
Hoen and Yahr stage number (%) Stage 2 27 (58.7) Stage 2.5 5 (10.9) Stage 3 12 (26.1) Stage 4 1 (2.2)	

Values are presented as mean ± SD for continuous characteristics.

**Table 2 ijerph-20-03806-t002:** Comparing clinical in-lab measurements pre intervention vs. immediate, within 1-week post intervention.

	Pre-Intervention	Post-Intervention	*p*-Value
**PD Related Symptoms**			
MDS-UPDRS III	29.76 (±13.37)	29.07 (±12.19)	0.500
LEDD (Mg/d)	602.71 (±303.39)	605.87 (±301.44)	0.622
NFOG-Q (total)	10.50 [0.00–20.25]	6.00 [0.00–17.00]	**0.007**
**Gait and Balance**			
Gait Speed (m/s)	1.14 (±3.2)	1.21 (±2.8)	0.101
Gait Speed DT (m/s)	0.91 (±2.8)	0.99 (±2.4)	**0.016**
10MWT (sec)	8.67 [7.78–10.62]	8.45 [6.95–10.05]	** 0.034 **
10MWT_DT (sec)	10.65 [9.10–13.60]	10.28 [8.70–12.00]	**0.040**
MiniBEST	19.00 [15.00–21.25]	21.00 [17.00–24.00]	**<0.001**
TUG (sec)	11.54 [8.82–15.00]	10.30 [7.84–12.45]	** 0.002 **
TUG_DT (sec)	16.87 [11.84–24.28]	14.27 [11.13–18.98]	**0.020**
30 Seconds Sit To Stand (Repetitions)	9.50 [7.00–11.25]	11.00 [7.75–12.25]	**0.006**
6MWT (Meters)	361.22 (±120.42)	406.83 (±11.58)	**<0.001**
Falls (Number/Month)	0.00 [0.00–0.20]	0.00 [0.00–0.00]	0.134
**Cognitive and mental health measurements**			
MoCA	22.20 (±3.78)	23.80 (±3.83)	**<0.001**
FES-I	24.50 [21.00–35.00]	25.00 [19.75–32.25]	0.159
GDS	4.50 [1.00–5.00]	2.00 [0.00–14.00]	**0.005**
SF-12 (health)	13.70 (±2.46)	14.33 (±2.21)	0.065
SF-12 (mental)	20.00 [17.00–23.00]	20.00 [17.00–22.25]	0.552
SF-12 (total)	33.39 (±5.37)	34.20 (±5.13)	0.185

Values are presented as mean ± SD for continuous variables with normal distribution; values are presented as median [IQR] for variables that were not normally l distributed or ordinal characteristics; normal distribution is based on Kolmogorov-Smirnov test; *p*-value is based on dependent t-test/Wilcoxon, *p*-value < 0.05; UPDRS-Unified Parkinson’s Disease Rating Scale; LEDD-Levodopa Equivalent Daily Dose; NFOG-Q-New Freezing of Gait Questionnaire; Gait Speed-10 m/seconds; Gait Speed Dual Task-10 m/seconds; 10MWT-10 m walk test; 10MWT_DT-10 m walk test dual task; MiniBEST-Mini Balance Evaluation Systems Test; TUG-Timed Up and Go Test; TUG_DT-Timed Up and Go Dual Task; 6MWT-6 min walk test; MoCA-Montreal Cognitive Assessment; FES-I-Falls Efficacy Scale International; GDS-Geriatric Depression Scale; SF-12 Health Survey.

**Table 3 ijerph-20-03806-t003:** Comparing clinical in-lab measurements pre intervention vs. 3-month follow-up after completion (n = 32).

Variable	Pre-Intervention	At 3-Months Follow-Up	*p*-Value
NFOG-Q (total)	17.50 [0.00–21.75]	8.50 [0.0–19.75]	**0.014**
**Dynamic balance and walking endurance**			
Gait Speed DT (m/s) **	0.92 (±2.8)	1.05 (±2.3)	** 0.004 **
10MWT (sec) *	8.67 [7.66–10.95]	8.12 [6.60–9.59]	**0.016**
10MWT_DT (sec) **	10.55 [9.02–13.60]	9.56 [8.10–11.80]	**0.007**
MiniBEST *	18.26 (±4.70)	20.03 (±4.26)	**0.018**
TUG (sec) *	12.70 [9.16–15.00]	9.87 [8.40–12.20]	**0.001**
TUG_DT (sec) *	18.10 [12.60–24.43]	15.30 [9.79–19.22]	**0.004**
30 Seconds Sit To Stand (Repetitions)	8.94 (±3.43)	9.72 (±4.66)	0.170
6MWT (Meters)	340.50 (±117.51)	387.25 (±133.39)	**0.012**
**General cognition and mental health**			
MoCA ***	22.58 (±3.51)	22.96 (±3.95)	0.563
Geriatric Depression Scale	5.00 [1.25–8.00]	3.50 [2.00–6.00]	0.212

Values are presented as mean ± SD for continuous variables with normal distribution and median [IQR] for variables with abnormal distribution or ordinal characteristics. A normal distribution is based on the Kolmogorov-Smirnov test. The *p*-Value is based on dependent t-test/Wilcoxon, *p*-value < 0.05. NFOG-Q-New Freezing of Gait Questionnaire; Gait Speed Dual Task-10 m/seconds; 10MWT-10 m walk test; 10MWT_DT-10 m walk test dual task; MiniBEST-Mini Balance Evaluation Systems Test; TUG-Timed Up And Go Test; TUG_DT-Timed Up And Go Dual Task; 6MWT-6 min walk test; MoCA-Montreal Cognitive Assessment; * Missing value = 1; ** Missing value = 2; *** Missing values = 8.

## Data Availability

Data is available upon reasonable request subject to compliance with local Helsinki committee guidelines.

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
