# Peer review of "Multidisciplinary Intensive Rehabilitation Program for People with Parkinson’s Disease: Gaps between the Clinic and Real-World Mobility"

_ijerph, 2023, doi:10.3390/ijerph20053806_

Round 1

Reviewer 1 Report

The work by Cohen et al. has assessed the effect of a multidisciplinary intensive rehabilitation program in improving daily-living walking quality and quantity in patients with Parkinson’s disease (PD). The manuscript is well-written, the methodological design and statistical approach seem appropriate and conclusions are supported by the data.

I only have two major comments for the authors:

1.- PD patients from this study are pretty old (mean age 71 years old). Could this influence in the final results? Do the authors believe that if the training program was adminisitered to younger PD subjects (or with lower disease duration) the results would be different? Can the authors perform some correlation analyses between pre-post differences in outcomes and disease duration?

2.- Line 69, only a subset of subjects was studied at 3-months. Those that completed the 3-month follow-up study are different from the subjects that did not complete it?

The remaining comments are minor:

3. The authors use the acronym PWPD to refer to people with PD. I highly recommend using PD acronym for referring to Parkinson’s disease throughout the text.

4. 3. Line 87, (MSA, PSP…) should go before “orthopedic”.

5. I detected several double-spaces after dots. For example, in lines 93, 115, 144, 182, 211, 270, 277, 280, 281.

6. UPDRS III and MoCA were performed by trained raters? If so, specify it in the text.

7. Line 155, has been the reproducibility of Axivity studied? Do the authors have any reference? If the accelerometer is not reproducible, comparing pre and post measurements does not make much sense, as this could be biased by the internal error of the test.

8. Line 163, please, describe what the abbreviations mean

9. Line 178, use PD instead of Parkinson’s disease

10. Line 189. A parenthesis “)” is missing

11. Line 189, performed instead of perfumed

12.  Line 190, a dot is missing at the end of the sentence

13. 13. MFOF-Q change it for MFOG-Q

14. Table 2. Specify in the title that this is 1-week post-intervention

15. Figure 1. Use bigger text in y and x axes. Also, specify if significant differences are present using * in the Figure. I would suggest using a “white boxplot” instead of “darkgrey” as the median value is not easily discernible with the current color.

16. Line 326, PD (use acronym).

17. In reference, extra spaces are present between ref 1 and 2, and ref 12 and 13

Author Response

Response to Reviewer 1 Comments

Revierwer 1 noted that the manuscript is well-written, the methodological design and statistical approach seem appropriate and conclusions are supported by the data.                                                     I only have two major comments for the authors:

Point 1:  a. PD patients from this study are pretty old (mean age 71 years old). Could this influence in the final results? Do the authors believe that if the training program was adminisitered to younger PD subjects or with lower disease duration) the results would be different?

  1. Can the authors perform some correlation analyses between pre-post differences in outcomes and disease duration?

Response 1:

  1. We thank the reviwer for this remark. Indeed, the cohort was comprised of relatively older and advanced people with Parkinson’s disease (PD). We agree that these facts might impact our findings and maybe with younger and less advanced patients the results might be different. This possibility was added to the revised version into the Discussion at the end of the first paragraph.
  2. As suggested, we performed a Spearman's correlation analysis of disease duration and the pre-post change in gait speed, step count, Timed up and Go test and the UPDRS part 3. Only the pre/post difference in the MiniBest was mildly correlated with age (Rs=0.292; p= 0.049). No other significant correlations were found (p> 0.127). These results are now included in the Results section below Table 3 and also briefly discussed at the end of the first paragraph of the Discussion.

Point 2: Line 69, only a subset of subjects was studied at 3-months. Those that completed the 3-month follow-up study are different from the subjects that did not complete it?  

Response 2:  Indeed, 32 participants were studied again about 3 month after completion of the 8 week intervention, (i.e., ~ 5 month from baseline testing). The subjects who were assessed at the 3-months follow-up and those who were not were similar at baseline. For example, age, sex, disease duration, UPDRS part 3, Hoehn & Yahr stage, MOCA, LEDD, gait speed, and Timed Up and Go were not different at baseline in those who completed the 3-months follow-up and those who did not (p>0.15). A sentence describing this informationwas added just above Table 3.

Minor comments:  

Point 3. The authors use the acronym PWPD to refer to people with PD. I highly recommend using PD acronym for referring to Parkinson’s disease throughout the text.

Response 3: The PD acronym was changed throughout the manuscript.

Point 4. Line 87, (MSA, PSP…) should go before “orthopedic”.

Response 4: We have done as suggested.

Point 5. I detected several double-spaces after dots. For example, in lines 93, 115, 144, 182, 211, 270, 277, 280, 281.

Response 5: Thank you for these observations. They have been corrected in the revised manuscript.

Point 6. UPDRS III and MoCA were performed by trained raters? If so, specify it in the text.

Response 6: Trained raters performed the tests. This information was added to the text in lines 145 and 151.

Point 7. Line 155, has been the reproducibility of Axivity studied? Do the authors have any reference? If the accelerometer is not reproducible, comparing pre and post measurements does not make much sense, as this could be biased by the internal error of the test.

Response 7: A number of studies have examined the properties and the validity of the measures derived from the Axivity in various populations including in patients with PD. Please see references below [1-5]. We added a clarifying sentence and these references into the text in section 2.3. (lines 185-189).  

Point 8. Line 163, please, describe what the abbreviations mean

Response 8:  SVM refers to the signal vector magnitude. This is in the text in line 163. We also added to the text in the methods: V=Vertical; ML= Medio/lateral; AP= antro/posterior.

Point 9. Line 178, use PD instead of Parkinson’s disease

Response 9: We have done as suggested.

Point 10 . Line 189. A parenthesis “)” is missing

Response 10: A parenthesis was added.

Point 11. Line 189, performed instead of perfumed

Response 11: This was corrected. Thank you for of all of these observations.

Point 12.  Line 190, a dot is missing at the end of the sentence

Response 12: A dot was added.

Point 13.  MFOF-Q change it for MFOG-Q

Response 13: This was changed into NFOG-Q.  

Point 14. Table 2. Specify in the title that this is 1-week post-intervention

Response 14: The title was revised accordingly.

Point 15. Figure 1. Use bigger text in y and x axes. Also, specify if significant differences are present using * in the Figure. I would suggest using a “white boxplot” instead of “darkgrey” as the median value is not easily discernible with the current color.

Response 15: Figure 1 was changed to take into account these constructive suggestions.

Point 16 . Line 326, PD (use acronym).

Response 16: Done.

Point 17. In reference, extra spaces are present between ref 1 and 2, and ref 12 and 13

Response 17: The extra spaces were removed.

Reviewer 2 Report

See PDF for comments.

Author Response

Response to Reviewer 2 Comments

Major Corrections:

Point 1. Definition of mobility.

Throughout the paper the authors refer to mobility but do not define this in the introduction.

There are multiple definitions of mobility and the outputs the authors describe in the results

allude to physical activity and physical activity behaviors. Please see the following papers from Fillekes et al and Cuignet et al which define mobility as a concept that describes an individuals movement in their environment both spatially and temporally encompassing factors such as the mode of transport and time out of home.

Fillekes MP, Giannouli E, Kim EK, Zijlstra W, Weibel R. Towards a comprehensive set of

GPS-based indicators reflecting the multidimensional nature of daily mobility for

applications in health and aging research. Int J Health Geogr. 2019 Jul 24;18(1):17. doi:

10.1186/s12942-019-0181-0. PMID: 31340812; PMCID: PMC6657041.

Cuignet, T., Perchoux, C., Caruso, G., Klein, O., Klein, S., Chaix, B., Kestens, Y., & Gerber,

  1. (2020). Mobility among older adults: Deconstructing the effects of motility and

movement on wellbeing. Urban Studies, 57(2), 383 401.

https://doi.org/10.1177/0042098019852033

Please clarify the definition of mobility and consider whether physical activity or mobility is

being captured.

Response 1: Thank you for this valuable remark. Indeed, after reviewing these two papers, we refer to the term mobility in the second paragraph of the Introduction. In addition, in the revised mansucript, we now cite these two articles.   

Point 2. Clarification on the breakdown of H&Y scores across the population.

  1. The stage of H&Y is intrinsically linked with mobility and therefore it should be discussed more

when drawing conclusions about the population as a whole. Please clarify the breakdown of participants by H&Y as opposed to just reporting H&Y stage 2.

  1. Additionally, the authors must detail the H&Y scores of the 32 participants that completed the 3

month post intervention assessments. This is essential in understanding whether the population

at completed the study i.e. completed the 3 month follow up is representative of the population that started the intervention.

Response 2:  a. We fully agree with this point. In the revised manuscript, as suggetsed, we have included the breakdown by H&Y stages.

  1. The H&Y stages of the 32 participants who completed the follow-up testing were similar to those who did not complete the follow-up testing, as now noted in the revised manuscript. Please see also the response to Reviewer 1, point 2 above.

Point 3. More detail is required on the method used to generate the daily living gait outcomes (line

number 154). In Section 2.3, it is simply stated that daily living step count, the percentage of the day the subject is active and mean signal vector magnitude were all generated. The authors must

provide more detail on the software package and methods used to generate these outputs.

Without this additional information it is impossible to understand whether these methods are

valid and accurate in this context. Additionally, when reporting the SVM calculation in line

number 163 the variables in the equation need to be stated. Please detail the methodology on accelerometer data so it is reproducible to the reader.

Response 3: As suggested, more details on the method used to generate the daily living gait outcomes and refferences were added to the text in Sections 2.3 and 2.4 and into Supplementary material.

Point 4. Clarification on qualitative measurements of daily-living gait. In Section 2.4, the authors refer to cadence, step length, gait speed, and stride and step regularity as qualitative measurements. It is unusual to refer to these outputs in this way as they are quantitative outputs. As stated previously, significantly more detail must be provided on values in SI units (m/s) instead of (cm/sec).

Please clarify the reasoning behind referring to the measurements of daily-living gait as

qualitative.

Response 4: Previous work has referred to the quantity of walking completed during the day (e.g., step count) and the quality of walking (e.g., step regularity). We have modified the text in the revised manuscript to more clearly explain this grouping and have also referred the reader to earlier work that used similar groups. We have changed the values into m/s instead of cm/s as suggested.

Point 5. Presentation of results.    The presentation of data in Table 1, especially anthropometric outcomes is very odd. It would be more appropriate to report height, weight and BMI as mean±sd as these are continuous variables. There are similar inconsistencies throughout the manuscript i.e. Table 2 SF-12 (health) is presented as mean±sd, SF-12 (mental) is presented as median[IQR], and SF-12 (total) is presented as mean±sd. As these outputs are all taken from the same questionnaire they should

be presented consistently. Please revise all tabulations accordingly as currently the interpretation of the data are greatly impeded by the current formatting.

Response 5: We agree with the remark that some outcomes are presented as mean and others as median. As suggested, we fixed Table 1 and age, height, and BMI are now repoarted as mean±SD. For outcomes, we use mean±SD if the measure was normally distributed and median and IQR otherwise. This is explanation is mentioned in the Material and Mmethods in the Statistical section 2.5 , and also was fixed under each table in the legend Normal distribution is based on Kolmogorov-Smirnov test." 

Point 6.  Clarification on the measurements carried out at each timepoint (line number 139).

At present, it is very difficult to understand when each clinic assessment was carried out and

also how many participants there were at each stage. To make this clear to the reader, the

authors should include a flowchart highlighting the number of participants and the

measurements taken pre intervention, 1 week post intervention, and 3 weeks post intervention.

Please include a step by step diagram, to support the methods section.

Response 6: Thank you for this valuable comment. As suggested, in the revised manuscript, we included a flowchart (new Figure 1) with this information to make it clearer to the readers. This was added to the Results section 3 at the end of the first paragraph. 

Point 7.  Immediate post intervention and 3 month post intervention

The authors do not present the differences between the outputs collected 1 week post

intervention and 3 months post intervention. This could provide context as to whether there is a 'wash out period' post intervention.

Response 7: With testing, testing after the intervention, and testing 3 months later, retention and washout can be inferred either by comparing both the values at both post-intervention time points to the baseline measures, and drawing the relevant inferences, or by comparing the post-intervention values to those 3 months later. We opted for the former.

Minor Corrections:

Point 1.  To improve the introduction the authors could provide some detail on how real world daily

living walking quantity or quality are currently measured (line number 58).

Response 1: As suggested some more details of how real world daily living walking are currently measured were added to the introduction along with propr refferences.

Point 2.  In line 75 please could the author clarfy the age of older adults.

Response 2:  We clarrified in line 75 that age ranged from 51 to 85 years old.

Point 3. Line number 157, please could the authors provide details on the type of skin tape used.

Response 3: The specific type of the skin tape was added to section 2.3.

Point 4. Line number 159, can the authors clarify if the recordings were kept if it was 3 days of 24hr

complete data or a specific number of data points each day.

Response 4: In Section 2.3, we now more clearly specify in the revised manuscript that at least 3 full days of Axivity recording were used, otherwise subjects were excluded from the analysis. 

Point 5. In Figure 1, in the top graph for responders the values on the y axis have been cropped so

20,000 reads 0,000.

Response 5: Thank you for this observation. This figure was corrected. Since we added a new flowchart as Figure 1 in the revised manuscript, this figure is now Figure 2.  

Point 6. The two graphs in Figure 1 could be combined onto one box and whisker plot to allow the

reader to easily compare the non-responders and responders.

Response 6: As suggested, in the new version of the manuscript, we combined the two parts into one figure.

Point 7. When reporting names of methods used the authors often do not reference these, examples

include Lee Silverman Voice Treatment (line number 112) and the type of SPSS package used

(line number 189).

Response 7: Thank you for this comment. In the revised manuscript we added a reference for the LSVT treatment (~line 114) and the type of the SPSS package in 2.5. Statistical Analyses.

Reviewer 3 Report

This study on 51 people with Parkinson’s disease (PD) explored the influence of an 8-week multidisciplinary intensive outpatient rehabilitation (MIOR) program on gait and balance, and measures of the quantity and quality of daily-living walking. Using common clinical assessment of function, the authors found that participants improved on most of the clinical assessments of function, which was maintained 3 months after the intervention. However, using a single wearable tri-axial accelerometer to assess daily-living walking, the authors found that none of the measures changed immediately or 3 months after the intervention. A secondary analysis of “responders” and “non-responders” based on post-intervention increased daily-living step counts, revealed the “responders” improved in their regularity and number of steps, whereas the non-responders did not change or got worse.

This study provides novel and clinically important information to the field of rehabilitation in PD. Comments are as follows:

·       Introduction:

o   Please include/expand upon the evidence/prior studies for what constitutes “improved” daily-walking.

o   Pg 1 lines 60-61: Please expand on and provide details of prior studies of lab-based gait and balance measures not reflecting measures of daily-life walking

·       Materials & Methods:

o   Pg 3 description of the MIOR program: Please provide more details on the amount of time spent within each 5-hour day. Having more exercise/activity dosing information will be helpful to readers.

o   Lines 104-105: For example, did participants also perform their “daily exercises” during the 8-week program, or was the development of those (home exercises) at the end of the 8-week program? Please also comment on how that may have impacted the 3-month follow-up; whether the “responders” followed these daily exercises with greater adherence than the “non-responders”.

·       Figure 1: The first plot vertical axis needs to be corrected.

o   Also, please clarify (add to legend) in this first plot the apparent outlier data point and the number “41) adjacent.

Author Response

Response to Reviewer 3 Comments

This stdy on 51 people with Parkinson’s disease (PD) explored the influence of an 8-week multidisciplinary intensive outpatient rehabilitation (MIOR) program on gait and balance, and measures of the quantity and quality of daily-living walking. Using common clinical assessment of function, the authors found that participants improved on most of the clinical assessments of function, which was maintained 3 months after the intervention. However, using a single wearable tri-axial accelerometer to assess daily-living walking, the authors found that none of the measures changed immediately or 3 months after the intervention. A secondary analysis of “responders” and “non-responders” based on post-intervention increased daily-living step counts, revealed the “responders” improved in their regularity and number of steps, whereas the non-responders did not change or got worse. This study provides novel and clinically important information to the field of rehabilitation in PD.

Comments are as follows:

Introduction:

Point 1: Please include/expand upon the evidence/prior studies for what constitutes “improved” daily-walking.

Response 1: A number of papers have described interventions that were designed to improve daily-living step counts. Some of these are now mentioned in the Introduction Less is known about how the spatial-temporal parameters of gait measured during daily-living change in response to an intervention. The introduction has been revised to make these ponts more clear. 

Point 2:     Pg 1 lines 60-61: Please expand on and provide details of prior studies of lab-based gait and balance measures not reflecting measures of daily-life walking 

Response 2: Thank you for this remark. In the revised version. we addedd several references and the below paragraph to the Introduction. "Accumulating work suggests that gait and balance, as measured using conventional testing in clinical and laboratory settings, does not necessarily reflect the parallel measures of gait during daily-living [11-16]. Giannouli et al. demonstrated only mild to moderate correlations between lab-based measured of spatial-temporal measures of gait and daily-living measures and multiple regression analyses indicated that the laboratory measures accounted for a significant but very low proportion of the variance (between 5% and 21%) of the corresponding daily-living measures. Similarly, Hillel et al. showed that in older adults, average daily-living gait speed is similar to that of dual-task gait speed in the lab. In addition, Shema-Shiratzky et al. reported that the typical values of daily-living gait speed and cadence were significantly lower than usual-walking values in the lab in both patients with multiple sclerosis and age-matched healthy controls. Fnally, Warmerdam et al. summarized studies that directly compared assessments made in unsupervised and supervised (e.g,, in the laboratory or hospital) settings, pointing to large disparities, even in the same parameters of mobility. These differences appear to be affected by psychological, physiological, cognitive, environmental, technical factors, and by the pathology. To facilitate the successful adaptation of the unsupervised assessment of mobility into clinical practice and clinical trials, clinicians and researchers should consider these disparities and the multiple factors that contribute to them."

Materials & Methods:

   Point 3: description of the MIOR program: Please provide more details on the amount of time spent within each 5-hour day. Having more exercise/activity dosing information will be helpful to readers.

   Response 3: We have added more details on the MIOR program including the amount of time spent within each 5-hour day and description of some of the activities. These details were added to section 2.1 ; The multidisciplinary intensive outpatient rehabilitation (MIOR) program.

   Point 4:   Lines 104-105: For example, did participants also perform their “daily exercises” during the 8-week program, or was the development of those (home exercises) at the end of the 8-week program? Please also comment on how that may have impacted the 3-month follow-up; whether the “responders” followed these daily exercises with greater adherence than the “non-responders”.

   Response 4:  All subjects received similar instructions regarding exercise and were reffered to centers in the community. The trainers, both the PT and the OT encourage the participants to maimtain physical activity, but no structured protocol of self-exercises was delivered. This was added in the end of section 2.1.  Still, it is possible that the responsders followed these instructions more carefully than the non-responders. This possibility is now mentioned in the Discussion of the revised manuscript.

   Point 5:    Figure 1: The first plot vertical axis needs to be corrected.

Response 5:  This was corrected.

Point 6:       Also, please clarify (add to legend) in this first plot the apparent outlier data point and the number “41) adjacent.

Response 6: In the legend of this figure in the revised mansucript, we now briefly discuss this outlier. 

Reference List

          1                    Del DS, Galna B, Godfrey A, Bekkers EMJ, Pelosin E, Nieuwhof F, Mirelman A, Hausdorff JM, Rochester L: Analysis of Free-Living Gait in Older Adults With and Without Parkinson's Disease and With and Without a History of Falls: Identifying Generic and Disease-Specific Characteristics. J Gerontol A Biol Sci Med Sci 2019;74:500-506.

          2                    Del DS, Kirk C, Yarnall AJ, Rochester L, Hausdorff JM: Body-Worn Sensors for Remote Monitoring of Parkinson's Disease Motor Symptoms: Vision, State of the Art, and Challenges Ahead  J Parkinsons Dis 2021;11:S35-S47.

          3                    Elshehabi M, Del DS, Hobert MA, Warmerdam E, Sunkel U, Schmitz-Hubsch T, Behncke LM, Heinzel S, Brockmann K, Metzger FG, Schlenstedt C, Rochester L, Hansen C, Berg D, Maetzler W: Walking parameters of older adults from a lower back inertial measurement unit, a 6-year longitudinal observational study. Front Aging Neurosci 2022;14:789220.

          4                    Hausdorff JM, Hillel I, Shustak S, Del DS, Bekkers EMJ, Pelosin E, Nieuwhof F, Rochester L, Mirelman A: Everyday Stepping Quantity and Quality Among Older Adult Fallers With and Without Mild Cognitive Impairment: Initial Evidence for New Motor Markers of Cognitive Deficits?  J Gerontol A Biol Sci Med Sci 2018;73:1078-1082.

          5                    Shema-Shiratzky S, Hillel I, Mirelman A, Regev K, Hsieh KL, Karni A, Devos H, Sosnoff JJ, Hausdorff JM: A wearable sensor identifies alterations in community ambulation in multiple sclerosis: contributors to real-world gait quality and physical activity.  J Neurol 2020;267:1912-1921.

Round 2

Reviewer 1 Report

The authors addressed all my previous concerns. 

I only have one final suggestion. The PWPD acronym was not changed in the Abstract. Please, use PD instead of PWPD also in the abstract.

Congratulations for you work.

Author Response

As suggested, this acronym is now changed in the Abstract.

Reviewer 2 Report

The authors have considered the majority of my previous comments and have addressed them. 

However, there are still some points that have not been addressed. 

1. Please can you provide the data for H&Y scale for participants that completed the follow up period. This is essential in understanding the population captured and whether H&Y score had a link to drop out rate. 

2. Please plot the new Figure 1 again. The non-responder and res0onders should be plotted on the sake y axis (step count) to allow the reader to directly compare. 

Author Response

  1. Please can you provide the data for H&Y scale for participants that completed the follow up period. This is essential in understanding the population captured and whether H&Y score had a link to drop out rate. 

RESPONSE: As suggested, this information has been added to the Results (after Table 1).

  1. Please plot the new Figure 1 again. The non-responder and responders should be plotted on the sake y axis (step count) to allow the reader to directly compare. 

RESPONSE: thank you for pointing this out. We prefer to use the default y-axis ranges from SPSS as this highlights the changes over time in each group. To address this point, in this new version of the manuscript, we note the differences in the y axis range in the legend to this figure.